# A Single-Cell Landscape of Spermioteleosis in Mice and Pigs

**DOI:** 10.3390/cells13070563

**Published:** 2024-03-22

**Authors:** Meng-Meng Liu, Chu-Qi Fan, Guo-Liang Zhang

**Affiliations:** College of Animal Science and Technology, Qingdao Agricultural University, Qingdao 266109, China; 20212103018@stu.qau.edu.cn (M.-M.L.); 630679356@qau.edu.cn (C.-Q.F.)

**Keywords:** single-cell transcriptome, mouse, pig, spermatogenesis

## Abstract

(1) Background: Spermatozoa acquired motility and matured in epididymis after production in the testis. However, there is still limited understanding of the specific characteristics of sperm development across different species. In this study, we employed a comprehensive approach to analyze cell compositions in both testicular and epididymal tissues, providing valuable insights into the changes occurring during meiosis and spermiogenesis in mouse and pig models. Additionally, we identified distinct gene expression signatures associated with various spermatogenic cell types. (2) Methods: To investigate the differences in spermatogenesis between mice and pigs, we constructed a single-cell RNA dataset. (3) Results: Our findings revealed notable differences in testicular cell clusters between these two species. Furthermore, distinct gene expression patterns were observed among epithelial cells from different regions of the epididymis. Interestingly, regional gene expression patterns were also identified within principal cell clusters of the mouse epididymis. Moreover, through analysing differentially expressed genes related to the epididymis in both mouse and pig models, we successfully identified potential marker genes associated with sperm development and maturation for each species studied. (4) Conclusions: This research presented a comprehensive single-cell landscape analysis of both testicular and epididymal tissues, shedding light on the intricate processes involved in spermatogenesis and sperm maturation, specifically within mouse and pig models.

## 1. Introduction

Spermatogenesis, the intricate process leading from spermatogonia to spermatozoa, has been a focal point of extensive research for more than a century [1]. The testis and epididymis play crucial roles in sperm transport, maturation, and storage. Sperm are initially produced in the testis, transported through the epididymis, and reach maturity within the epididymis. Despite attaining a specific morphological structure in the testis, spermatozoa remain functionally immature, necessitating further processing in the epididymis.

The epididymis, an indispensable component of the reproductive system, serves diverse functions, including sperm storage, secretion of nutritive fluid, facilitation of sperm development and maturation, and enhancement of sperm vitality [2]. Comprising three primary regions: the caput, corpus, and cauda, the epididymis facilitates further modifications of nonmotile sperm [3,4]. Within the epididymis, the sperm plasma membrane undergoes sequential biochemical and proteomic modifications due to interactions with components of the extracellular environment within the epididymal lumen [5]. During this process, sperm are progressively concentrated as they travel from the caput to the corpus and ultimately to the cauda epididymis, where they can be stored in an inactive state for extended periods [6]. Each epididymis region exhibits a distinct gene expression pattern crucial for specific physiological functions during various stages of sperm maturation [7,8,9,10].

While previous large-scale transcriptomic investigations have focused on bulk RNA from aggregates of multiple spermatogenic cell types [11,12], recent advances in high-throughput single-cell or single-nucleus RNA-sequencing technologies have enabled in-depth examinations of cellular and molecular aspects of the testis [13,14] and epididymis. Previous studies mainly provided a concise overview and specific gene expression of mouse spermatogenesis and sperm mutation. There was also a report on the somatic cell changes that occur during spermatogenesis in pig testis [15]. In contrast to previous studies that solely focused on either spermatogenesis or sperm maturation, this study comprehensively explores both processes. However, a comprehensive understanding of spermatogenesis and sperm maturation across mouse and pig lineages is still lacking. Animal models play a key role in scientific research. Mice have a high degree of genetic purity and high fertility. Therefore, the mouse is a commonly used model in mammal-related research [16]. As rodents, mice are different from domestic animals to some extent. Therefore, we included pigs as a contrast model. First, the pig is the most common domestic animal in China. In addition, the pig is more human-like than the mouse in terms of both anatomy and physiology, making them another attractive option for simulating humans [17]. Therefore, we used mice as model animals to elucidate the mechanism of spermatogenesis and maturation and used pigs as livestock models to further study spermatogenesis and maturation.

Widespread translational regulation of transcriptomes occurs during spermatogenesis within the testis [18]. The liquid microenvironment in the epididymal duct, with its ionic components exhibiting segmental specificity, provides a platform for sperm motility and maturation. Numerous studies have shown that the expression of innate immune secretory genes in epididymal epithelial cells plays a crucial role in spermatogenesis [19,20,21,22,23,24]. Additionally, epididymal β-defensin family proteins have been confirmed to be essential for host reproductive tract defense and spermiogenesis [23,24,25].

We employed single-cell RNA-seq to identify comprehensive gene expression patterns in individual cells from testis and epididymis samples from mice and pigs. These data were validated by protein immunostaining using testis and epididymis tissue. Additionally, we identified subpopulations of testicular cells and investigated their gene expression profiles. *ELAVL2* and *CCNB2* were found to be potential markers for specific spermatogenesis processes. Furthermore, we characterized different segments of the mouse and pig epididymides, describing their unique gene expression characteristics. Next, we explored the types of principal cells and segmental specific expression of β-defensin family genes in the mouse epididymis. The results revealed a conserved yet dynamic continuum of gene expression patterns across the full spectrum of spermatogenic development, as well as heterogeneity indicative of spermatogenic cell subtypes engaged in distinct biological pathways or functions. Our analysis of the data revealed different species and lineage-specific cellular and molecular features of spermatogenesis and sperm maturation in different species.

## 2. Materials and Methods

### 2.1. Animals

Sample collection was conducted under license in accordance with the Guidelines for Care and Use of Laboratory Animals of China, with the approval of the Institutional Review Board of Qingdao Agricultural University. The 8-week-old male C57BL/6J mice used in our study were purchased from Beijing Vital River Laboratory Animal Technology Co., Ltd. (no. 11002009000012, production license number: SCXK: 2023-7077, Beijing, China). All animals were maintained under conditions of ad libitum access to water and food with constant light-dark cycles. For single-cell and transplant studies, testes and epididymides from 8-week-old male C57BL/6J mice (at least two per experiment), both from Beijing Vital River Laboratory, were used to generate suspensions of cells following enzymatic digestion, as previously described [26]. Six 12-month-old Landrace pigs were obtained from Haiyang Hexing Animal Husbandry Co., Ltd. (no. 11002009000017, production license number: SCXK: 2023-9065, Yantai, China).

### 2.2. Sample Preparation for Single-Cell RNA-Sequencing

#### 2.2.1. Generation of Epididymis Cell Suspensions

For both mice and pigs, the epididymides were dissected and divided into three regions (caput, corpus, and cauda) as previously described [7]. To obtain a sperm-depleted single-cell suspension, epididymis samples were dissected into a 10 cm plate containing 10 mL of Krebs media prewarmed to 35 °C. Once the dissected samples were cleared of any visible sperm, 3 mL of the sample mixture was transferred to a small 25 mL glass Erlenmeyer flask containing 7 mL of dissociation media (Collagenase IV 4 mg/mL, Deoxyribonuclease I (DNAse I) 0.05 mg/mL) (Servicebio, Wuhan, China) and placed in a 35 °C water bath with rotation at 200 rpm for 30 to 45 min. Afterwards, the samples were transferred to a conical tube and allowed to settle. The supernatant was removed, leaving 4–5 mL of sample solution to be returned to the Erlenmeyer flask, where 10 mL of 0.25% trypsin Ethylenediaminetetraacetic acid (EDTA) (Servicebio, Wuhan, China) and 0.05 mg/mL DNAse I were added prior to the return to the water bath shaker. After 35 min, the samples were pipetted up and down until there were no observable pieces of tissue, and allowed to incubate in a water bath for an additional 20 min. Once cell disaggregation was achieved, 1 mL of fetal bovine serum (FBS) (Servicebio, Wuhan, China) was added to inactivate the trypsin. The sample was transferred to a Falcon tube while passing the cell solution through a series of 100 mm, 70 mm, and 40 mm cell strainers, and media supplemented with 10% FBS and antibiotic and antimycotic (100 units/mL penicillin, 100 mg/mL streptomycin, and 0.025 mg/mL amphotericin B) (Servicebio, Wuhan, China) were added to a final volume of 30 mL. Finally, the samples were subjected to centrifugation for 15 min at 400 relative centrifugal force (*g*) at room temperature (RT). The supernatant was discarded, and up to 25 mL of media was added prior to centrifugation for 5 min at 800 relative centrifugal force (*g*) at RT. Afterwards, the supernatant was discarded, 2 mL of phosphate buffered saline (PBS) was added, and the cells were transferred to a 2 mL microfuge tube. The media were further removed by centrifugation at 900 relative centrifugal force (*g*) for 3 min, the supernatant was discarded, and the pellet was resuspended in 600 mL of PBS.

#### 2.2.2. Generation of Testis Cell Suspensions

The testis from mice and pigs were cleaned and washed with PBS. These samples were stored in MACS Tissue Storage Solution within 48 h (Miltenyi Biotec, Bergisch Gladbach, Germany). Before dissociation, the testis tissues were cut into small pieces and transferred to 0.2% collagenase IV and DNase I digestion solution, followed by incubation at 37 °C for 15 min. After digestion and mechanical striking into single cells, the cell suspension was filtered through a cell strainer and converted to barcoded scRNA-seq libraries in accordance with the manufacturer’s protocol. The sequencing libraries were sequenced by DNBSEQ-T7 (MGI, Shenzhen, China).

### 2.3. Analyses of Single-Cell Transcriptomes

Raw count matrices generated by Cell Ranger were imported to Seurat 4.3.0 and filtered for only high-quality cells. Briefly, we removed cells with fewer than 200 detected genes and genes detected in 3 or fewer cells. Gene expression values were log normalized and scaled before further downstream analyses. Cell clustering and uniform manifold approximation and projection (UMAP) analysis were performed based on the statistically significant principal components. Marker genes of each cell cluster were determined by a log fold change threshold above 0.25 using the default Wilcoxon rank-sum test. After the identification of cell clusters, the raw count matrices of the data subsets were imported to Monocle and only genes with expression above the threshold (0.1) were used for the analyses. Differentially expressed genes or significantly variable genes among cells were identified by Monocle and used for ordering cells in pseudotime. Only genes with a dispersion ratio above 0.1 were used for training the pseudotime trajectories. To generate the pseudotime heatmaps, differentially expressed genes (DEGs) among cell clusters in pseudotime with qval <0.1 were included and clustered hierarchically based on their expression trends.

### 2.4. Visualization

In Seurat, the uniform manifold approximation and projection (UMAP) technique was used to map high-dimensional cellular data into a two-dimensional space, bringing together cells with similar expression patterns and separating cells with different expression patterns further apart. The differences between the cells were thus made more comprehensible.

### 2.5. Testis Tissue Immunostaining

*ELEVL2* (Rabbit pAb (A5918, ABclonal, Wuhan, China)) and *CCNB2* (Cyclin B2 Rabbit mAb (A7956, ABclonal, Wuhan, China)) were subjected to immunofluorescence staining experiments. Mouse and pig samples were collected and washed three times with PBS, fixed in 4% paraformaldehyde, embedded in paraffin, and sliced into 5 μm sections. These sections were then held at 6 5 °C for 1 h, deparaffinized, and rehydrated. The sections were finally incubated in citrate antigen retrieval solution (pH 6.0) at 96 °C for 10 min. As the sections gradually cooled, they were blocked in QuickBlock™ Blocking Buffer for Immunostaining (Beyotime, Nantong, China) for 10 min and incubated with primary antibodies at 4 °C overnight. Then, the sections were incubated with secondary antibodies for 30 min at 37 °C, washed with PBST after incubation with antibodies, and incubated with Antifade Mounting Medium containing DAPI (Beyotime, Nantong, China).

### 2.6. Mouse Epididymis Fluorescence In Situ Hybridization

The mouse epididymal paraffin section was incubated at 62 °C for 2 h, followed by xylene dewaxing for 30 min. Subsequently, the section was treated with anhydrous ethanol for 20 min and washed with PBS for 10 min. The slices were repaired using citric acid antigen retrieval solution (ServiceBio, Wuhan, China) at a temperature of 96 °C for 10 min and allowed to cool naturally to room temperature. A digestion step was performed by adding protease K (ServiceBio, Wuhan, China) at a concentration of 20 µg/mL in a volume of 100 μL, followed by incubation at 37 °C for 10 min and subsequent washing with PBS three times. Prehybridization solution (100 μL) was added and incubated at a temperature of 37 °C for 1 h. Then, hybridization solution containing the RNA probe Defb20 (5‘UTR: CATCTGAGTGCCAAAGTTCTAAAACATCTTGGGCTGCTTAACATCTGGGCTCTAATCTGGCCTAATCTGCTTCTTCAG), Defb30 (5′UTR: AAGAGCACGAGGGTCAACTGGCACTGGTAGGGAGGAGAGCAGCAGGTGTAAATCCGTTTTTTCATGTGACTGATG) was added (60 μL). Incubation overnight took place at a temperature of 37 °C followed by washing with saline sodium citrate (SSC). For signal detection, the hybrid solution containing the signal probe (ServiceBio, Wuhan, China) was added in a volume of 60 μL and incubated at a temperature of 37 °C for 1 h before being washed again with SSC. Finally, nuclei were labeled using 4′,6-diamino-2-phenylindole (DAPI; Beyotime, Jiangsu, China), observed under a laser scanning confocal microscope (Hitachi, Tokyo, Japan).

### 2.7. Ethical Statements

All experimental procedures involving animals were approved by the Qingdao Agricultural University Institutional Animal Care and Use Committee (DEC2024-0119, 17 March 2024).

### 2.8. Data Availability

The raw sequence data reported in this paper have been deposited in the Genome Sequence Archive [27] in National Genomics Data Center [28], China National Center for Bioinformation/Beijing Institute of Genomics, Chinese Academy of Sciences (GSE249819), which are publicly accessible at https://ngdc.cncb.ac.cn/gsa (accessed on 1 February 2024). Additional information related to the data in this study is available from the lead contact upon request.

## 3. Results

### 3.1. Transcriptional Atlas of the Testis and Epididymis in Mice and Pigs

To systematically investigate spermatogenesis in the testis and sperm maturation in the epididymides of mice and pigs, we prepared single-cell suspensions from testicular and epididymal tissue and performed single-cell RNA-seq using the DNBelab C4 platform. After low-quality cells were removed, 45,639 cells were obtained from the testis and epididymis joint data of the mouse. On average, we detected 41,960 genes expressed in each individual cell. For the pig testis and epididymis joint data, we profiled 42,712 individual cells. On average, 20,011 genes expressed in individual pig cells were detected (Appendix A). To analyze the dataset, we performed UMAP on the combined datasets using the Seurat package. There were more clusters or heterogeneous cell types in the mouse samples (12 clusters) than in the pig samples (10 clusters). In the mouse UMAP, clusters 1, 2, 5, 8, and 10 were testis cells, and clusters 0, 3, 4, 6, 7, 9, and 11 were epididymis cells (Figure 1A). Pig UMAP showed that clusters 4–6 were testis cells, and clusters 0–3 and 7–9 were epididymis cells (Figure 1B). In addition, the top 10 differentially expressed genes (DEGs) were identified in each of the cell clusters using the Find Markers function in Seurat, and the top 10 DEGs in both mice and pigs are shown in heatmaps (Figure 1C,D). Interestingly, both mouse and pig testicular cells were continuous and clearly separate from the epididymal cells, as indicated by the degree of dispersion in the UMAP landscapes (Figure 1A,B).

To better elucidate the mechanism of spermatogenesis in the testes of mice and pigs, we analyzed testicular tissue from the two groups. Specifically, we profiled 15,555 single-cell testes from mice and detected 36,943 genes expressed in single-cell testes on average. A total of 6831 single cells from pig testis samples were analyzed, with an average of 17,079 genes expressed in each single cell (Appendix A). Different gene expression profiles and UMAP clusters were detected in mice and pigs (Figure 2). This suggested that there are significant differences in gene expression between mice and pigs. Then, we aimed to further elucidate sperm post testicular maturation. We extracted mouse and pig epididymides from three regions: the caput, corpus, and cauda. We profiled 41,012 genes across 30,307 cells in the mouse epididymis, and 32,575 genes spanning 15,618 cells in the pig epididymis (Appendix A).

### 3.2. The Single-Cell Atlas of the Testis in Mice and Pigs Shows That Spermatogenesis Is a Continuous Process

To identify mouse testis cell clusters via uniform manifold approximation and projection (UMAP), we projected the expression of sets of known cell type-specific marker genes (Appendix A). The quality-filtered mouse testis cells were identified at different stages as 15 different germ cells along with somatic cells including undifferentiated spermatogonia (Id4+), differentiating spermatogonia (Sohlh1+ and Stra8+), spermatocytes (Sycp1+ and Sycp3+), spermatids (Acrv1+, Catsper3+ and Catsper4+), fibroblasts (Col1a1+), macrophages (Cd68+ and Cd74+), endothelial cells (Vwf+, Eng+ and Cdh5+), Sertoli cells (Clu+ and Wt1+), and Leydig cells (Insl3+, Cyp11a1+ and Star+) (Figure 2A). Unexpectedly, there was an unknown cluster that was not identified for a certain cell type. To address the molecular and functional characteristics of these cells, we identified DEGs using Gene Ontology (GO) analysis (Figure 2C). The GO terms associated with the DEGs in this cell cluster included the top 20 terms related to spermatogenesis and sperm motility (Appendix A). Here, our computational analysis suggested the occurrence of continuous spermatogenesis in mice. Following the direction of the arrow, there were spermatogonial clusters, spermatocyte clusters, and spermatid clusters (Figure 2A).

As with mouse testis, pig testis analysis yielded 11 major clusters or cell types (Figure 2B) that were subsequently annotated using known marker genes (Appendix A). These cells included undifferentiated spermatogonia (ID4+, ZBTB16+), differentiating spermatogonia (KIT+, DMRT1+), spermatocytes (SYCP1+, SYCP3+), and spermatids (ACRV1+, CATSPER3+, CATSPER4+): somatic cells, such as Leydig cells (STAR+, INSL3+); endothelial cells (ENG+, CDH5+, CD34+); macrophages (AIF1+, CD68+); fibroblasts (COL1A1+); and Sertoli cells (SOX9+, GATA4+, WT1+). According to the order of spermatogenesis, a continuous biological process occurs from spermatogonia through spermatocytes to spermatids (Figure 2B). By comparing the testicular cells of the two species, we found that their somatic and germ cell types were similar, but they had different germ cell subtypes, as distinguished by differentially expressed genes. It is important to note that the mice have additional testis cell types and numbers than pig. In addition, a greater number of genes were also detected in mouse testicular cells. Overall, the mechanisms of spermatogenesis might differ between mice and pigs.

### 3.3. Gene Expression Patterns of Testes in Mice and Pigs

To determine the specific class of genes associated with the differences between mouse and pig testes, we investigated the contribution of different gene categories to the overall UMAP. Here, we visualized several representative markers of mouse and pig testes (Appendix A). The Acrv1 (ACRV1) gene was significantly expressed in mouse and pig spermatids (Appendix A). There are many typical genes that are expressed differentially between the two species. Stra8 was specifically expressed in mouse differentiating spermatogonia (Appendix A). In contrast, it was barely expressed in pig testis cells (Appendix A). Another differentiating spermatogonial gene, Sohlh1, which is also expressed in Aal, and A1–A4, is expressed at intermediate levels; it is expressed at low levels in B spermatogonia [29]. Sohlh1 is specifically expressed in differentiated mouse spermatogonia but was not detected in pigs (Appendix A). The expression of the CCNB2 gene was like that of SYCP family genes in single-cell samples of pig testes (Appendix A). Leydig cells were defined by Star and Insl3 and macrophages specifically expressing the genes for Cd68 and Cd74 in mouse. Similarly, we identified pig Leydig cells with the marker genes STAR and INSL3 (Appendix A).

### 3.4. Developmental Trajectory-Related Differences between Mice and Pigs during Spermatogenesis Development

We applied Monocle2 to infer the pseudotemporal trajectory of cells from a cohort of germ cells involved in the spermatogenesis. Based on our UMAP clustering analysis, we selected undifferentiated spermatogonia, differentiating spermatogonia, spermatocytes1, and spermatids2 clusters from mice to generate germ cell fate trajectories. In pigs, the cluster of undifferentiated spermatogonia contains a fraction of Sertoli cells. Hence, this cluster was excluded. We selected differentiating spermatogonial, spermatocytes1, spermatocytes2, and spermatids3 clusters to perform a pig trajectory study. Unbiased dynamic cell trajectory analysis of the pseudotime order with spermatogonia from both mice and pigs (Appendix A) yielded continuous development. The mouse trajectory yielded eleven developmental hierarchies (State 1–11), while the pig trajectory yielded five developmental hierarchies (State 1–5).

### 3.5. Spermatogonia Development Trajectory and Gene Expression Profiles between Mice and Pigs

To further analyze the pseudotime differences in spermatogenesis between the two species, we extracted the trajectory of germ cell development in spermatogonial clusters from mice and pigs (Figure 3A,B). Specifically, the development trajectory analysis of mouse spermatogonia included a total of 1555 single cells, involving undifferentiated spermatogonial clusters and differentiated spermatogonial clusters, while the development trajectory analysis of pig differentiating spermatogonia included only 536 single cells. A more systematic analysis via heatmap and clustering provided a format to explore and display the gene ontology (GO) terms as well as the magnitude of genes that showed dynamic expression along with the germ cell differentiation timeline (Appendix A). Projecting the mouse spermatogonia onto the development trajectory revealed three stages in the timeline (from state 1 to state 3 to state 2). We observed that the first state of these cells was predominantly composed of genes specifically expressed in differentiating spermatogonia, such as Stra8, Sohlh1, Dmrt, and Rhox13 (Appendix A). Rhox13 has been confirmed to be required for quantitatively normal first-wave spermatogenesis in mice [30]. The next developmental stage (state 3) along the developmental trajectory corresponds to the undifferentiated spermatogonial marker Id4 and the additional increase in the expression of genes involved in cell cycle activation and control, such as Cox8c, Tex101, Pixil1, and Tesmin. The subsequent cluster (state 2) was enriched in biological processes such as ribonucleoprotein complex biogenesis and mRNA processing (Appendix A). Based on the gene expression patterns and GO heatmap (Appendix A), we assigned state 1 as a mouse that sporadically differentiated during the first wave of spermatogenesis. State 2 was related to the meiosis process. State 3 included some of the cells that provide energy and raw materials for mitosis.

Coincidently, in the trajectory analysis of the development of differentiating spermatogonia in pig testes, the GO term “meiosis process” was evident at the midpoint of the trajectory of the development of differentiating spermatogonia in pig testes (state 3). At the end of the pesudotime trajectory, the process of mitosis began to increase (state 6) (Appendix A). This suggested that the filtered spermatogonia data in mice and pigs both had two distinct types at the end of the trajectory, one undergoing mitosis to increase its number and the other undergoing meiosis to initiate spermatogenesis.

### 3.6. Different Dynamic Gene Expression Patterns of Spermatogonia in Mice and Pigs

Monocle provides a convenient way to screen all pseudotime-dependent genes and identify genes following similar kinetic trends. Furthermore, we identified gene expression signatures associated with distinct developmental stages. The heterogeneity of gene dynamics in mouse and pig spermatogonia was compared. For mouse spermatogonia, we examined the developmental trajectory expression of classical genes such as Id4, Piwil1, Sohlh1, and Stra8 (Appendix A). As shown in the figure, Stra8 and Sohlh1 were expressed only at the beginning of spermatogenesis [31]. KIT, a marker for differentiating spermatogonia [32] is directly regulated by Sohlh1 [33]. Interestingly, the developmental trajectory of Piwil1 expression in mice was similar to that of KIT in pig [34] (Appendix A). The expression levels of these genes were elevated in the middle trajectory. In mouse spermatogonia, when Sohlh1 expression ended, Piwil1 expression immediately increased (Appendix A). This is likely because of the potential relationship between the two genes. We also visualized several classic differentially expressed genes that exhibited dynamic expression during the mouse spermatogonial trajectory, such as Dkkl1 and Rbakdn. Their expression levels gradually increased along the developmental trajectory (Appendix A). A recently published study showed that DKKL1 may play a pivotal role in testicular development and spermatogenesis [35]. Rbakdn is specifically expressed in the mouse testis, and its expression level continued to increase after the meiosis stage [36].

For pigs, we visualized the differentiating spermatogonial marker genes DMRT1 and KIT [37] (Appendix A). The expression levels of these genes gradually increased over time, and after reaching a peak, their expression levels gradually decreased. A high expression level corresponds to a cell state in which meiosis is enriched. However, the expression trend for ART3 was exactly the opposite (Appendix A). It has been reported that ART3 expression is related to the quantitative impairment of spermatogenesis [38]. TEX101 was only expressed at the beginning of the developmental trajectory and was also named state 1 (Appendix A). Previous studies have shown that TEX101 is expressed only in germ cells and is thought to be involved in spermatogenesis in adult male mice [39]. This showed that the expression pattern of these genes was closely related to their function. For DEGs related to the trajectory of pig spermatogonial development, we visualized ELAVL2 and UCHL1 (Figure 3C). ELAVL2 plays posttranscriptional roles in the regulation of spermatogonial proliferation and apoptosis [40]. UCHL1 in the testis can induce germ cell apoptosis and control factors that protect against apoptosis during spermatogenesis [41].

KIT, DMRT1, and ELAVL2 were highly expressed in different spermatogonial clusters; however, the development trajectory of ELAVL2 was obviously different. However, the expression of ELAVL2 increased in the late stage of development (state 6), which was enriched in the mitotic process (Figure 3C). We were particularly interested in ELAVL2 in spermatogonial cells from mice and pigs. To investigate the extent of synchrony, or lack thereof, between gene expression at the RNA and protein levels and to validate our single-cell RNA-seq profiles at the protein level, we performed IF (immunofluorescence) staining of the ELAVL2 gene. Immunofluorescence staining also showed that ELAVL2 was abundantly distributed in mouse and pig spermatogonia. As shown in the figure, DDX4, which is an RNA helicase expressed in the germ cells of all animals [42]^,^ and ELAVL2 were double stained in mouse and pig testes, respectively. Costaining of DDX4 and ELAVL2 further highlighted these differences between spermatogonia and other germ cells (Figure 3D and Appendix A).

### 3.7. Developmental Differences between Mouse and Pig Spermatocyte Clusters

After establishing the developmental pathways of spermatogonial cells, we integrated the scRNA-seq dataset with available clusters of spermatocytes. These clusters had an atlas of 2504 single cells from mice and an atlas of 1597 single cells from pigs. Notably, both mouse and pig developmental trajectory trends were similar and contained five states (Figure 4A,B). Pseudotime analysis allowed us to reconstruct the developmental pathways involved in spermatocyte development, including gene dynamic expression and biological process enrichment. Along the spermatocyte developmental trajectory, we compared the enriched biological processes between mice and pigs. In mouse spermatocyte pseudotime GO terms (Appendix A), biological processes such as the regulation of nuclear division, axoneme assembly, and plasma membrane bound cell projection assembly were enriched in the first stage (Appendix A). Similarly, in the first stage, GO terms for pig spermatocytes (Appendix A), multicellular organismal reproductive process, cell maturation, and regulation of the cell cycle were enriched (Appendix A). Next, we examined the GO analysis of mouse state 3 along the developmental trajectory. Synapse organization, male gametogenesis, and assembly reproduction processes were enriched in this group. The corresponding state in pig spermatocytes was state 2, which was enriched in secretory granule organization, spermatid differentiation, and cytoplasmic translation. The last state of the mouse spermatocyte developmental trajectory was state 4, in which protein localization to the mitochondrial membrane, chromosome condensation, and DNA packaging processes were established and enriched. In the pig spermatocyte developmental trajectory, the last state was state 3, where spindle organization, cell cycle processes, and Golgi vesicle transport were enriched. In both mouse and pig spermatocytes, biological processes associated with meiosis were highly enriched.

### 3.8. Dynamic Gene Expression Patterns between Mice and Pigs

To link developmental stage and gene transcription, we analyzed highly expressed genes in developmental trajectory cells from the mouse and pig spermatocytes. Moreover, we detected the expression of multiple genes involved in processes unique to spermatocytes. The expression levels of the spermatocyte marker genes Sycp2 and Sycp3 [43] were found to be high in initial spermatocytes, but decreased gradually in later stages in mice (Appendix A). The projection of the expression levels of SYCP family genes in pigs was in stark contrast to that in mice. The expression levels of these genes gradually increased in the later stages (Appendix A). The spermatid marker gene Acrv1 was highly expressed in mouse spermatocytes at developmental trajectory state 4 (Appendix A) [44] and pig spermatocytes at developmental trajectory state 3 (Appendix A). It has been reported that the Acrv1 promoter is loaded with RNA II polymerase but paused in spermatocytes, thus ensuring precise transcriptional elongation in round spermatids [45]. Hence, Acrv1 expression in early spermatocytes is restrained but gradually increases in the late stage of spermatocytes. Another typical spermatid marker gene, Catsper4, is also expressed in mice in the spermatid stage 4. CATSPER3 was expressed in the last stage of spermatocytes in pigs. We also plotted the classical developmental trajectory DEGs of mouse and pig spermatocytes and visualized them in plots (Appendix A and Figure 4C). Lyar was highly expressed in all mouse spermatocytes (Appendix A), which is consistent with previous research showing that the protein is present in spermatocytes [46]. The potential mouse spermatocyte marker Ccna1 also had a high degree of similarity with the Sycp3 gene in terms of developmental trajectory (Appendix A). The Ccna1 protein encoded by this gene belongs to the highly conserved cyclin family, whose members are characterized by a dramatic periodicity in protein abundance throughout the cell cycle [47]. Thus, we deduced that Ccna1 might be a suitable marker of spermatocytes in mice. Among the DEGs related to the developmental trajectory of pig spermatocytes, CCNB2 and NKE2 were shown to be dynamically expressed (Figure 4C). The expression level of these genes peaked in state 1 and then gradually decreased. Studies in mouse testes have shown that Nek2 may play an important role during mitosis and meiosis in mice and pigs [48]. CCNB2 is a crucial gene required for cells to enter the M phase during mitosis and meiosis [49]. As anticipated, high expression of CCNB2 was observed in pig spermatocytes. To refine our analysis, we examined the expression of this protein in mouse and pig testes by CCNB2 immunofluorescence (Figure 4D). Our study demonstrated the expression of CCNB2 in early-stage spermatocytes and epithelial cells, as well as spermatogonia in pigs. Additionally, our findings revealed the presence of CCNB2 protein in Leydig cells in mice, which aligns with the developmental trajectory analysis. Therefore, CCNB2 could serve as a valuable marker for identifying Leydig cells in mice.

### 3.9. Unique Haploid Transcriptome Facilitates Spermiogenesis

Mouse and pig steady state spermatogenic cell clusters (Appendix A) expressing spermatid genes (mouse cluster spermatids2; pig cluster spermatids3) were further resolved via developmental trajectory analysis based on DEG patterns in each species. There were 1348 single cells in the mouse spermatid developmental trajectory and 1295 single cells in the pig spermatid developmental trajectory. Their developmental trajectory analysis revealed a common pool of progenitor cells that subsequently separated into two branches (Appendix A). We removed the branches and analyzed only the main line in the spermatid developmental trajectory. Each branch was divided into three clusters to construct a GO heatmap (Appendix A). Along with mouse spermatid developmental trajectory sequencing, there were three states: state1, state2, and state4 (Appendix A). The dynamic expression pattern of genes within clusters and the developmental trajectory indicate the potential role of biological processes. Intriguingly, genes enriched in mouse spermatids were related to centriole replication and purine nucleotide catabolic processes in the early stages according to GO analysis (Appendix A), positive regulation of protein processing, and chromosome condensation upregulation during the latter half of the trajectory (Appendix A). According to the GO terminology, these cells undergo a process that begins with metabolic processes and ends with sperm metamorphosis in mouse spermatids. The developmental trajectory of pig spermatids begins with chromosome condensation, and spermatid development ends with protein localization and other sperm deformation processes (Appendix A).

### 3.10. Different Dynamic Gene Expression Patterns between Mouse and Pig Spermatids

To further explore the molecular differences underlying mouse and pig spermiogenesis, gene expression data or spermatids were extracted from the scRNA-seq data and combined for detailed developmental trajectory analysis. In mice, we identified specific cohorts of genes that exhibited maximal expression during the spermatid trajectory. Gsg1, which is predicted to enable RNA polymerase binding activity and is located in the endoplasmic reticulum membrane, was highly expressed in mouse spermatids (Appendix A). Spata19 was highly expressed at stage 4 in the mouse spermatid developmental trajectory (Appendix A). Our spermatid developmental trajectory analysis is consistent with a prior study showing that Spata19 plays an important role in sperm motility by regulating the organization and function of mitochondria [50]. Surprisingly, we analyzed Defb family genes in state 1, such as Defb19 and Defb36 (Appendix A). β-Defensins are small antimicrobial peptides that play essential roles in male fertility [51]. The majority of β-defensins are primarily expressed in the male reproductive tract and play roles in sperm maturation and capacitation [52]. Defb19 is involved in the migration of germ cells [53]. The abundant expression of Defb genes in mouse testes protects against microbial invasion. Immunostaining revealed enrichment of Defb30 and Defb20 in the mouse epididymis (Appendix A).

Similarly, in mice, the expression of GSG1 was also remarkable in the pig spermatid developmental trajectory (Appendix A). A previous study suggested that GSG1 plays a role late in meiosis following DNA replication [54]. The results indicated that spermatid cell clusters in mice and pigs were mixed with cells in meiosis anaphase. The HMGB4 gene was reported to be present in the euchromatin of late pachytene spermatocytes and haploid round spermatids during spermatogenesis. However, it is preferentially expressed in mouse testes and is localized to the basal pole of elongated spermatids [55]. Among our DEGs to the developmental trajectory of pig spermatids, HMGB4 was highly expressed in the late stages of spermatogenesis (Appendix A). PHF7, a key factor in sperm chromatin aggregation [56], was highly expressed in the developmental trajectory of pig spermatids and had a trend similar to that of HMGB4 (Appendix A). Taken together, we identified cell type(s) represented in each cluster, including major spermatogenic cell types, spermatogonia, spermatocytes, and spermatids, by cell type-specific gene expression (Figure 5).

### 3.11. Single-Cell Transcriptomes of the Epididymis in Mice and Pigs

We performed single-cell RNA-sequencing on epididymal cells from mice and pigs. The mouse and pig epididymids were divided into three functional regions: the head (caput), body (corpus), and tail (cauda). Single-cell suspensions of the caput, corpus, and cauda of mice and pigs were prepared, respectively. First, integrative clustering of adult mouse epididymis scRNA-seq datasets was performed to identify and assign each cell to its correct subgroup. After batch effect correction, a total of 30,307 single cells and 41,012 features were annotated and separated into multiple clusters using UMAP (Figure 6A). Eleven cell clusters were defined according to the expression levels of specific markers including epithelial cells (Col1a1+), endothelial cells (Vwf+), fibroblasts (Col1a1+), monocytes (Csf1r+, Lyz2), basal cells (Krt8+), and principal cells (Cst11+, Lcn5+, Gpx3+). In the pig epididymis, there were a total of 32,575 features spanning 15,618 single cells. We constructed a single-cell atlas of the pig epididymis and defined thirteen cell clusters. Specifically, cell clusters were annotated as basal cells (KRT8+, KRT14+), monocytes (CSF1R+), myoid cells (ACTA2+), endothelial cells (VWF+), and principal cells (RNASE10+, CST11+, LCN5+, GPX3+). There are also several cell clusters that had not been clearly identified as cell types and these clusters were mostly derived from the cauda of the pig epididymis (Figure 6B).

### 3.12. Distinctive Principal Cell Programs across the Epididymis in Mice and Pigs

According to the marker genes of different epididymal regions, we found multiple discrete groups of principal cells arrayed along the mouse epididymis (Figure 6A), which is consistent with the findings of a previous study [6]. Regional expression of epididymal principal cells has been demonstrated in mice but not in pigs. In contrast to the principal cells of the epididymis in mice, the marker genes for the principal cells of the epididymis in pigs is the subject of relatively few genome-wide studies. Similarly to those in the mouse epididymis, principal cells in the pig epididymis were also detected using caput principal cell marker genes such as RNASE10, CST11, LCN2, and MFGE8; corpus principal cell marker genes such as LCN5, RNASE9, and PLAC8; and cauda principal cell marker genes such as AQP3, NOV, GPX3, and CRISP1 (Appendix A). Conversely, only caput and corpus principal cells were found in the pig epididymis (Figure 6B).

We aimed to gain further insights into the gene transcription of epithelial cells across the epididymis. We identified differentially expressed genes (DEGs) using the Find Markers function in Seurat. Analyses of genes conserved in the mouse epididymis and pig epididymis for each cell cluster were also performed (Appendix A). As shown in the gene expression map, the basal cell marker gene Krt8 (KRT8) was also highly expressed in the principal cells of both mice and pigs. KRT8 plays a role in maintaining cellular structural integrity and functions in signal transduction and cellular differentiation [57,58]. Basal cells can regulate electrolyte and water transport via principal cells [59]. Therefore, KRT8 (Krt8) may play an important role in transduction between principal cells and basal cells. The genetic mechanism of this signaling pathway in principal cells and basal cells is worthy of further investigation. Studies over the past 15 years have shown that β-defensins are abundantly expressed in the postnatal epididymides of different species [24,60,61]. The most notable feature of region-specific gene regulation in the epididymis is the segmental expression of individual members of large multigene families that are organized in genomic clusters, including β-defensins. Here, we projected UMAP of highly expressed Defb genes in mice and pigs. Interestingly, these genes were highly expressed in both mouse and pig epididymis principal cells (Appendix A). For example, Defb28 was upregulated in the mouse cauda principal cluster, and Defb30 was highly expressed in the mouse corpus principal cluster. In the pig caput epididymis, DEFB128 was highly expressed. In the pig corpus epididymis, DEFB129 was highly expressed (Appendix A). To further validate our single-cell RNA-sequencing (scRNA-seq) results, we performed immunostaining to visualize the products of representative Defb genes in the mouse epididymis. The mouse epididymis expressed higher levels of Defb20 in the caput and of Defb30 in the corpus according to the present data (Figure 7A,B). To confirm this, we performed fluorescence in situ hybridization (FISH) of certain exons of Defb20 and Defb30 in the mouse epididymis. Specifically, the FISH results showed that Defb30 was expressed throughout the epididymis but was expressed at the highest level in the corpus epididymis (Figure 7C–E). Compared with other regions, defb20 was highly expressed in the mouse caput epididymis (Figure 7F,G). Curiously, we found that defb20 and defb30 ended up decorating the sperm surface, which is also consistent with prior studies [6].

## 4. Discussion

The process of spermatogenesis occurs within the testes: spermatogonia, spermatocytes, and spermatids, after which the sperm mature into epididymis. To discern the cell composition and gene-expression dynamics of the tissues, we utilized 10× Genomics single-cell transcriptome technology to investigate the cell types in the testis and epididymis of mouse and pig samples, along with their gene expression profiles. This enabled us to compare the different gene expression patterns involved in spermatogenesis and sperm maturation between the two species. Five cell clusters were identified in the testis, and seven cell clusters were identified in the epididymis of mice. In pig samples, only three clusters were observed in the testis and seven clusters were observed in the epididymis. The mouse samples exhibited a greater diversity of cell types. It was evident that the testicular cells of both the mouse and pig samples were continuous and distinct from the epididymal cells. Furthermore, there are significant differences in gene expression patterns during sperm development and maturation between mice and pigs. Moreover, a more detailed examination of the differences in spermatogenesis between mouse and pig samples was conducted by further analysis.

Spermatogenesis is recognized as a complex process [62,63]. In contrast to previous single-cell analyses on mice [6,64,65,66] and pigs [67], the current data provide a comprehensive exposition elucidating the dynamic modifications occurring on sperm from the testis to the epididymis. Expanding upon prior research, this study offers an intricate examination of gene expression during spermatogenesis and sperm maturation in mice and pigs, while also identifying novel marker genes. Subsequently, we conducted a thorough analysis of gene expression during spermatogenesis in the testes of mice and pigs. We identified sequential gene expression profiles corresponding to established cell types or subtypes within the spermatogenic lineage, ranging from spermatogonia to spermatids, in mice and pigs, respectively. Notably, 15 cell clusters were detected in the mouse testis, while only 11 cell clusters were identified in the pig testis. Our findings indicated that mice possess more subtypes than pigs. For instance, mouse spermatocytes exhibit three subtypes, and spermatids display five subtypes, whereas pig spermatocytes possess two subtypes and spermatids have three subtypes. This suggests that the process of spermatogenesis in mice may be more complicated than in pigs, which agrees with previous studies [64]. From the perspective of gene lineage expression, significant disparities in spermatogenesis between mice and pigs were observed. *SOHLH1* was detected in mouse spermatogonia but not in pig spermatogonia, based on our data. This could indicate that the spermatogonia of mice are primarily type A intermediate spermatogonia, while those of pigs are predominantly type B spermatogonia [68]. In pig spermatogonia, the expression of *ELAVL2*, a little-studied *ELAVL* family member [69], is particularly active. Interestingly, the significantly differentially expressed genes identified in the pseudotime analysis of pig spermatogonia indicated that *ELAVL2* was highly expressed in pig spermatogonia. Consequently, immunostaining for *ELAVL2* was performed on mouse and pig testes. *DDX4*, a marker for germ cells, was used to separate germ cells from somatic cells. Costaining of *DDX4* and *ELAVL2* further highlighted the expression of *ELAVL2* in spermatogonia.

We detected the expression of meiosis-related genes in spermatocytes, including Dmrt*1*, *Sycp3*, *Tex101*, and *Catsper4*, which are important for the fertility of mice [70]. Interestingly, we detected the expression of Sertoli cell marker genes such as *SOX9, GATA4*, and *WT1* in pig undifferentiated spermatogonia. The results indicated that these types of cells had many genes with similar expression in pigs. Tadokoro et al. reported that Sertoli cells express the cytokine glial cell line-derived neurotrophic factor (GDNF) under the control of follicle-stimulating hormone, which is required for the maintenance of the undifferentiated spermatogonial population in vivo [71]. However, little is known about the relationship between Sertoli cells and undifferentiated spermatogonia in pigs. Future studies are needed to acquire more knowledge in this respect. Our GO analysis indicated that spermatogonia exhibit two different cell characteristics in both mice and pigs, one characteristic increases in number, and the others starts spermatogenesis.

Although the process of spermatogenesis is similar in mice and pigs, there are slight differences in gene expression patterns at each stage. *Dkkl1* and *Rbakdn* were significantly expressed in mouse spermatogonia but not in pig spermatogonia. In the meiosis-enriched stage of pig spermatogonia, *DMRT1* and *KIT* were significantly expressed, which differed from what was observed in the mouse stage. The developmental trajectory analysis revealed that the trajectories of spermatocyte development in mice and pigs were remarkably similar, but the enriched GO pathways of spermatocytes in mice were different from those in pigs. Developmental trajectory gene expression analysis indicated that *Defb* family genes were highly expressed during the initial stage of spermatids in mice. Chromosome condensation-related genes were upregulated at late stages of spermatids in pigs, while they were active in earlier stages of spermatids in mice. In summary, these results indicated that the process of spermatogenesis may be similar between mice and pigs, but gene expression and biological function enrichment greatly differed at each stage. Previous studies have shown that cyclin B2 (*Ccnb2*) can be detected from the spermatogonial stage to the spermatocyte stage in reptiles [72]. *Ccnb2* mRNA was also detected in spermatocytes of mice [73]. In the present study, we found that *CCNB2* was typically expressed in pig spermatocytes. Moreover, *CCNB2* is involved in the mitosis stages of spermatogonia [74]. Combined with the expression of the *CCNB2* gene in pig testes, our findings indicated that the *CCNB2* gene could be used as the potential marker gene for pig early spermatogenesis and could be identified by immunostaining. Taken together, our IF staining results confirmed the key markers identified through the transcriptome approach. Gene regional expression was visualized in the mouse epididymis. This is the basis for creating different microenvironments for sperm maturation. In addition, we explored the cell types of the epididymis that exhibited distinct differences between mice and pigs. Gene regional expression patterns were examined in mouse principal cells from our data. It is worth noting that compared with the testis, the epididymis shows “immune privilege”. The immune environment not only protects spermatozoa from autoimmunity but also defends spermatozoa against pathogenic damage [75]. In this study, we examined the expression of certain immune-related *Defb* families in the mouse and pig epididymides. This finding is consistent with the in situ hybridization of *Defb20* and *Defb30* in the mouse epididymis. It has been reported that *Defb20* is expressed specifically in the caput of the mouse epididymis and has the potential to be involved in the epididymal sperm maturation process [76]. *Defb30* is abundantly expressed in the male reproductive tract, where it most likely protects against microbial invasion [52]. These genes work together to promote sperm maturation and ensure normal fertilization in mice. In the pig epididymis, *DEFB128* and *DEFB129* were significantly detected in the principal cells of the caput and corpus, respectively.

Overall, cell types in the epididymis of mice and pigs showed obvious differences. A total of 11 cell type clusters were identified in the mouse epididymis, and 13 clusters of cell types were identified in the pig epididymis. The greatest difference between mice and pigs is that mouse epididymal principal cells show obvious regional gene expression patterns. Furthermore, no epididymal principal cells were detected in the cauda of the pig epididymis. The microenvironment of the cauda epididymis is thought to be essential for maintaining the fertilization ability of sperm [77]. The difference in the cell composition of the cauda between mice and pigs indicated differences in the functions of the different species. In the caput principal cells of the mouse epididymis, the marker genes included Rnase10, *Cst11*, *Lcn2*, and *Mfge8*, which are the same as the marker genes of pig principal cells in the caput. Cystatin 11 (*CST11*) belongs to the cystatin type 2 family of cysteine protease inhibitors and exhibits antimicrobial activity in vitro. This gene is regulated by androgen and encodes an epididymal-specific protein that has shown to have antimicrobial activity against *E. coli* [78,79]. Furthermore, lipocalin (*LCN*) family members are small, secreted proteins that bind to small hydrophobic molecules through their characteristic central beta barrel. Among them, *LCN2* has been reported to regulate insulin sensitivity and nutrient metabolism. In addition, *LCN2* deficiency in mice leads to increased susceptibility to bacterial infections [80]. Moreover, it had been reported that the binding of *MFGE8* to integrin α&β5 promotes termination of the insulin receptor signaling pathway in mice [81]. *RNASE10* has been identified as the most abundantly secreted protein in the proximal pig epididymis and plays a significant role in post testicular sperm maturation and male fertility [82]. It is clear that caput principal cells are closely related to insulin regulation and antimicrobial activity. In the corpus principal cells of the mouse epididymis, the marker genes were *Lcn5*, *Rnase9*, and *Plac8*. In pigs, the corpus principal cell marker genes were *RNASE9*, *PLAC8*, *CRISP1*, *AQP3*, and *NOV. Rnase9* and *Lcn5* were found to be directly or indirectly regulated by androgens [83,84]. In the caudal principal cells of mice, marker genes such as *Gpx3*, *Gstm2*, *Spink10*, and *Crisp1* were significantly expressed, and these marker genes were not detected in the caudal principal cells of pigs. The *CRISP1* gene encodes a multifunctional protein that is secreted into the epididymal lumen. The protein binds to the postacrosomal region of the sperm head, playing important roles in spermioteleosis [85]. In addition, the *CRISP1* gene can regulate the Ca^2+^ channels of sperm [86].The expression of *Gpx3* is regulated by the androgen epididymis of mice [87].

## 5. Conclusions

In summary, the single-cell transcriptome data presented an extensive, publicly accessible resource that significantly contributes to sperm maturation and storage, particularly for artificial pig insemination. The unsupervised pseudotime transition from spermatogonia to spermatids reflects the developmental trajectory of mouse and pig spermatogenesis. Although gene expression patterns during spermatogenesis in mice and pigs are somewhat similar, there are also species-specific differences. Additionally, we investigated other cellular components within the epididymis, such as fibroblasts and monocytes, and characterized their distribution and gene expression patterns in the mouse and pig epididymides. As a result, we obtained a comprehensive landscape of sperm development in both the testis and epididymis, which provides a basis for better understanding spermatogenesis and spermioteleosis.

## Figures and Tables

**Figure 1 cells-13-00563-f001:**
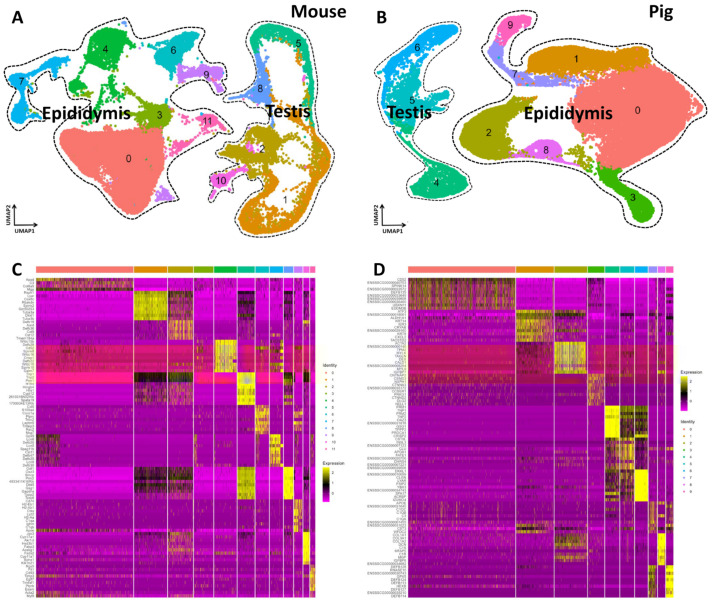
Single-Cell RNA Profiling of the Testis and Epididymis Between Adult Mice and Pigs Reveals the Extent of Gene Expression Heterogeneity. (**A**) Dimension reduction analysis (via UMAP) of combined single-cell transcriptome data showing single-cell RNA profiling of mice epididymis and testes. Each dot represents a single cell and is colored according to its cell type identity. (**B**) Dimension reduction representation (via UMAP) of combined single-cell transcriptome data showing single-cell RNA profiling of pigs’ epididymis testes. Each dot represents a single cell and is colored according to its cell type identity. (**C**) Heatmaps showing the top 10 significantly differentially expressed genes (DEGs) between each cell cluster in mice testes and epididymis samples. (**D**) Heatmaps showing the top 10 significantly differentially expressed genes (DEGs) between each cell cluster in pigs’ testes and epididymis samples.

**Figure 2 cells-13-00563-f002:**
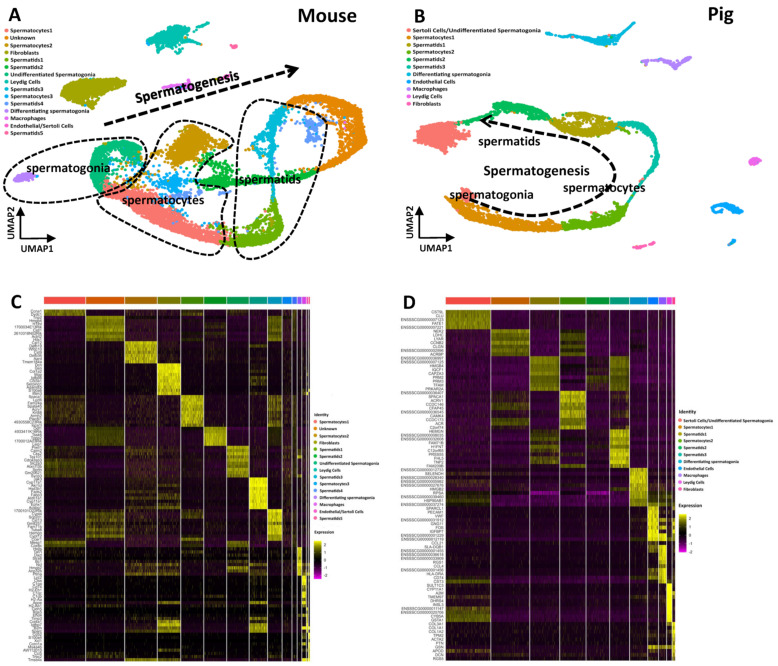
Single-Cell Transcriptomes of Mice and Pigs Testicular Cells. (**A**) Dimension reduction representation (via UMAP) of the mice testes single-cell transcriptome. Each dot represents a single cell and is colored according to its cell type identity. The right clusters is spermatogonia, the middle clusters is spermatogonia, and the left contains spermatids clusters. Along the direction of the arrow, is the sequence of spermatogenesis (**B**) Dimension reduction representation (via UMAP) of the pigs’ testes single-cell transcriptome. Each dot represents a single cell and is colored according to its cell type identity. The arrow starts with the spermatogonia cluster and the spermatocyte clusters in the middle, and the spermatids clusters at the end. Along the direction of the arrow, is the sequence of spermatogenesis (**C**) Heatmaps show the top 10 differentially expressed genes (DEGs) between each cell cluster in mice testes. (**D**) Heatmaps show the top 10 differentially expressed genes (DEGs) between each cell cluster in pigs’ testes.

**Figure 3 cells-13-00563-f003:**
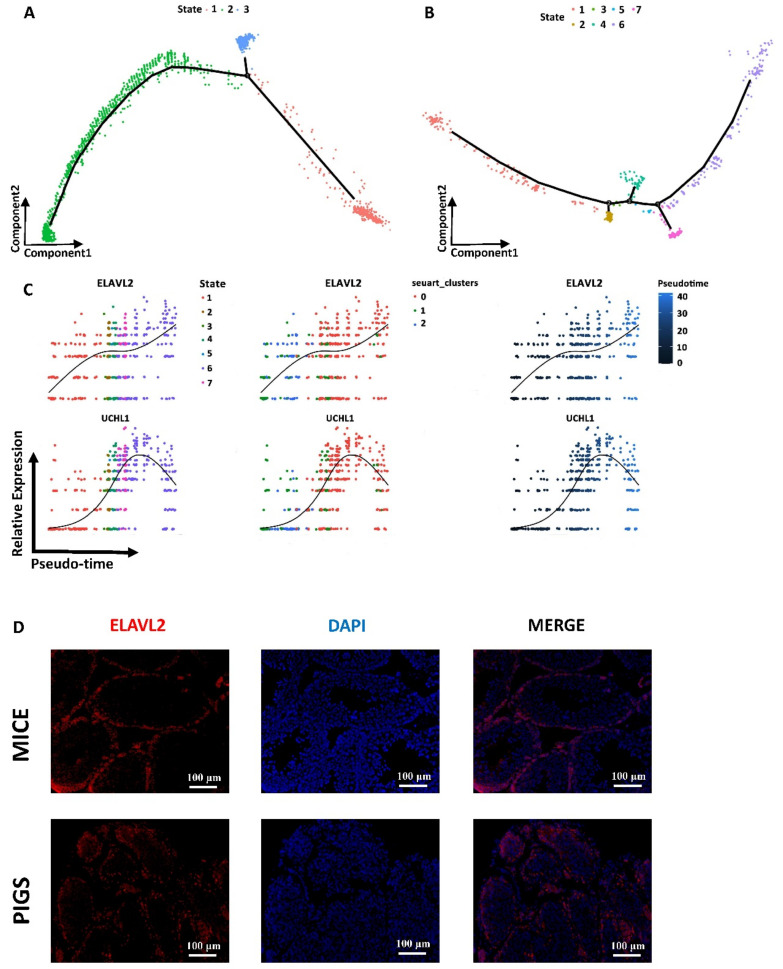
Single-Cell Spermatogonial Trajectories Reveal Heterogeneity Between Mice and Pigs. (**A**) Pseudotime trajectories of mice spermatogonia in which cells are colored by state. Branch points in the single-cell trajectories are noted by black numbered circles. The spermatogonia clusters in this trajectory analysis included undifferentiated spermatogonia and differentiating spermatogonia clusters. There are 3 states in mouse spermatogonia pseudotime trajectories. (**B**) Pseudotime trajectories of pigs’ spermatogonia in which cells are colored by state. Branch points in the single-cell trajectories are noted by black numbered circles. The spermatogonia clusters included in this trajectory analysis included differentiating spermatogonia cluster. There are 7 states in pigs spermatogonia pseudotime trajectories. (**C**) Expression patterns of key DEGs over pseudotime among pigs’ spermatogonia in state, cluster, and pseudotime. Cells are colored according to the state, clusters, and pseudotime and ordered according to the pseudotime. The left is DEGs in different states’ dynamic changes along pigs spermatogonia pseudotime trajectories. The middle is DEGs in different seurat-clusters’ dynamic changes along pigs spermatogonia pseudotime trajectories. The right is DEGs’ dynamic changes along pigs spermatogonia pseudotime trajectories. (**D**) Red indicates immunostaining for ELAVL2 and blue indicates staining for DAPI in mice and pigs’ testes (the bar represents 100 μm). Scale bars are indicated. DNA was counterstained with DAPI. A minimum of three animal samples were used for each genotype and each experiment was repeated three times with similar results.

**Figure 4 cells-13-00563-f004:**
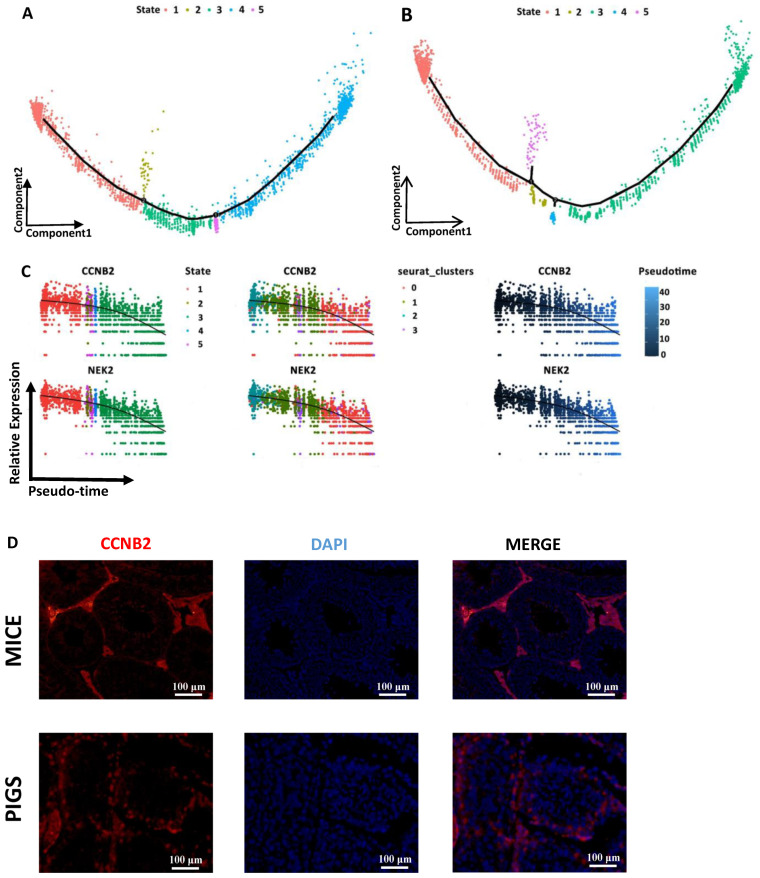
Dynamic Transcriptomic Heterogeneity in Mice and Pigs Spermatocytes. (**A**) Pseudotime trajectories of mice spermatocyte in which cells are colored by state. Branch points in the single-cell trajectories are noted by black numbered circles. Spermatocyte clusters included in this trajectory analysis included the spermatocytes1 cluster. There are 5 states in mice spermatocytes pseudotime trajectories. (**B**) Pseudotime trajectories of pigs’ spermatocytes in which cells are colored by state. Branch points in the single-cell trajectories are noted by black numbered circles. The spermatocyte clusters included in this trajectory analysis included spermatocytes1 and spermatocytes2 clusters. There are 5 states in pigs’ spermatocytes pseudotime trajectories. (**C**) Expression patterns of key DEGs over pseudotime among pigs’ spermatocytes in state, cluster, and pseudotime. Cells are colored according to the state, clusters, and pseudotime and ordered according to the pseudotime. The left is DEGs in different states’ dynamic changes along pigs’ spermatocytes pseudotime trajectories. The middle is DEGs in different seurat-clusters’ dynamic changes along pigs’ spermatocytes pseudotime trajectories. The right is DEGs’ dynamic changes along pigs’ spermatocytes pseudotime trajectories. (**D**) Red immunostaining for CCNB2 and blue DAPI staining of mice and pigs’ testes (the bar represents 100 μm). Scale bars are indicated. DNA was counterstained with DAPI. A minimum of three animal samples were used for each genotype and each experiment was repeated three times with similar results.

**Figure 5 cells-13-00563-f005:**
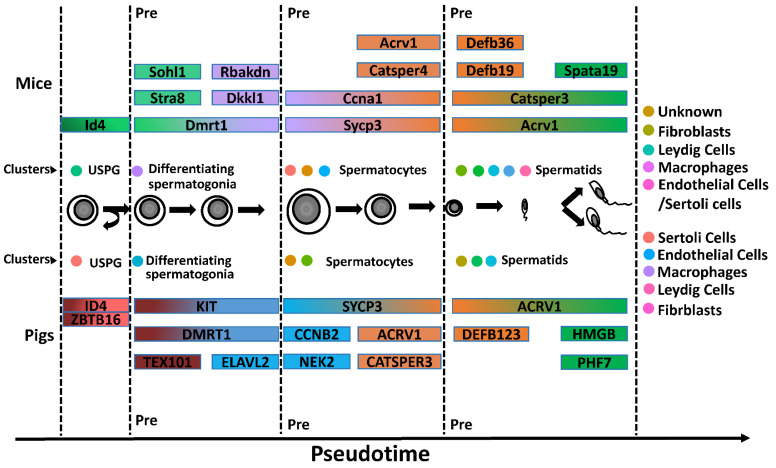
Comparison of Dynamic Transcriptomes of the Testis between Mice and Pigs. Identification of cell clusters expressing the noted marker genes allowed clusters to be aligned with specific spermatogenic cell types. From spermatogonia to spermatocytes and finally to spermatids in mice and pigs, respectively, each dot represents the corresponding cell cluster, showing the differentially expressed and highly expressed marker genes at each stage (USPG: undifferentiated spermatogonia).

**Figure 6 cells-13-00563-f006:**
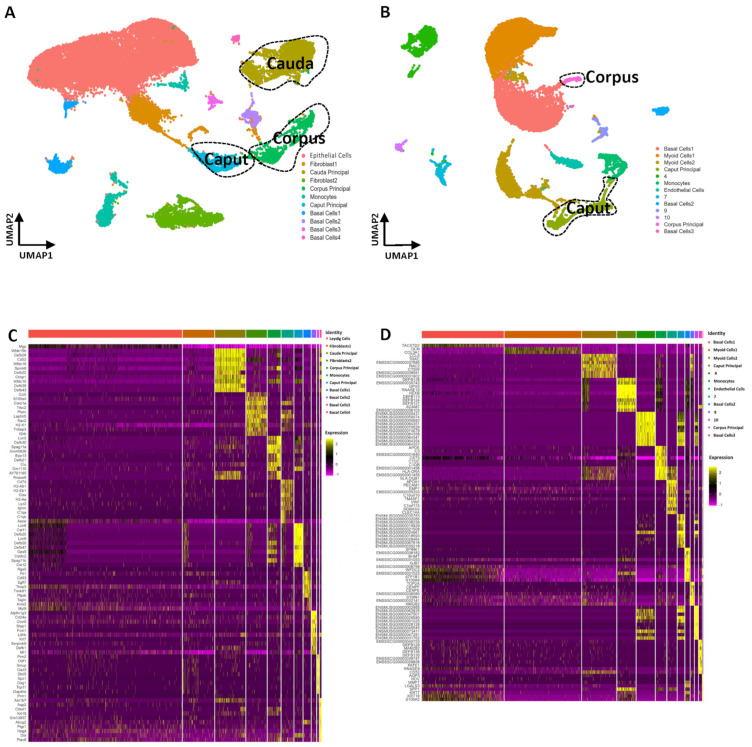
Single-Cell Transcriptomes of the Epididymis of Mice and Pigs. (**A**) Dimension reduction map (via UMAP) of the mice epididymis single-cell transcriptome showing the regional expression of specific principal cells. Each dot represents a single cell and is colored according to its cell type identity. (**B**) Dimension reduction presentation (via UMAP) of the pigs’ epididymis single-cell transcriptome. Each dot represents a single cell and is colored according to its cell type identity. (**C**) Heatmaps showing the top 10 differentially expressed genes (DEGs) between each cell cluster in the mice epididymis. (**D**) Heatmaps showing the top 10 differentially expressed genes (DEGs) between each cell cluster in the pigs’ epididymis.

**Figure 7 cells-13-00563-f007:**
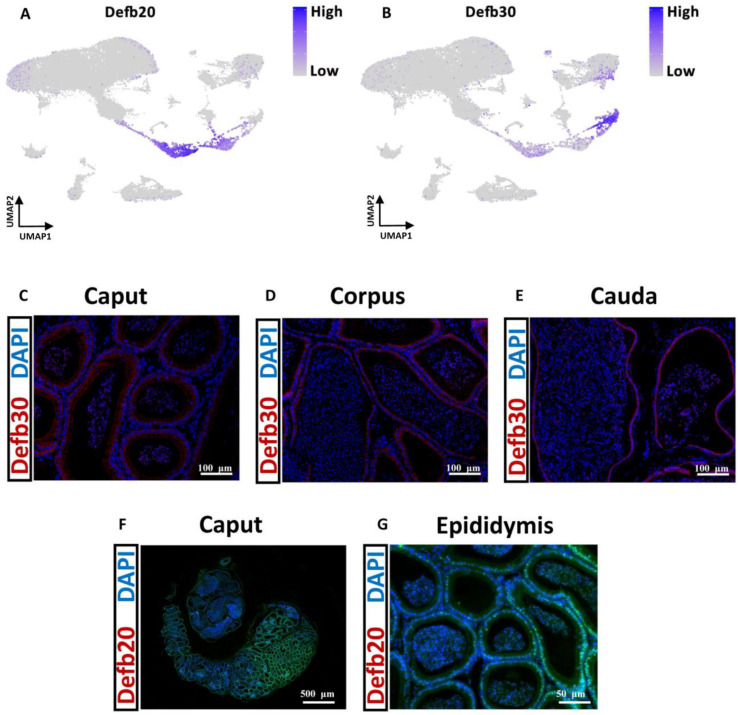
Localization of Dfb20 and Defb30 in the Mouse Epididymis. (**A**) Expression patterns of Defb20 projected on the UMAP plot in mouse epididymis. The darker the blue color is, the greater the expression level. (**B**) Expression patterns of Defb30 projected on the UMAP plot in mouse epididymis. The darker the blue color is, the greater the expression level. (**C**) Defb30 in situ hybridization of the mouse caput epididymis (the bar represents 100 μm). (**D**) Defb30 in situ hybridization of the mouse corpus epididymis (the bar represents 100 μm). (**E**) Defb30 in situ hybridization of the mouse cauda epididymis (the bar represents 100 μm). (**F**) Defb20 in situ hybridization of the mouse caput epididymis (the bar represents 500 μm). (**G**) Defb20 in situ hybridization of the mouse epididymis (the bar represents 50 μm). Scale bars are indicated. A minimum of three animal samples were used for each genotype and each experiment was repeated three times with similar results.

## Data Availability

All data supporting our findings are included in the manuscript.

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
