# Peer review of "A Single-Cell Landscape of Spermioteleosis in Mice and Pigs"

_cells, 2024, doi:10.3390/cells13070563_

Round 1

Reviewer 1 Report

Comments and Suggestions for Authors

The authors should clarify the purpose/advantages of this comparison between the mouse and the pig.  There is a large amount of information in the data presentation but the comparison to previously published literature should indicate what this contribution adds to the field that is not already known.  Technical aspects and data analysis is very good. Purpose less so.

Comments on the Quality of English Language

Editing by an authority is required.  That is not my job as a reviewer other than to point out that most of the problems are associated with the use of an incorrect tense.

Author Response

Question 1:The authors should clarify the purpose/advantages of this comparison between the mouse and the pig.  There is a large amount of information in the data presentation but the comparison to previously published literature should indicate what this contribution adds to the field that is not already known.  Technical aspects and data analysis is very good. Purpose less so.

Reply: Thank you very much for your kind words about our work. We think this is an excellent suggestion. The purpose/advantages of the mouse and the pig have been added to the manuscript in line 56 to line 64. The details are as follows: “Animal models play a key role in scientific research. Mouse has high degree of genetic purity as well as high fertility. Therefore, mouse is the commonly used model in mammal related research [1]. As rodents, mouse is different from domestic animals to some extent. Therefore, we induced pig as a contrast model. First, pig is the most common domestic animal in China. Besides, pig is more human-like, in both of anatomy and physiology, than mice, making them another attractive option for simulating human [2]. Therefore, we used mouse as model animals to elucidate the mechanism of spermatogenesis and maturation and adopted pig as the livestock model to further study spermatogenesis and maturation.”

[1]Rosenthal N, Brown S. The mouse ascending: perspectives for human-disease models. Nat Cell Biol. 2007 Sep;9(9):993-9. doi: 10.1038/ncb437. PMID: 17762889.

[2]Hou N, Du X, Wu S. Advances in pig models of human diseases. Animal Model Exp Med. 2022 Apr;5(2):141-152. doi: 10.1002/ame2.12223. Epub 2022 Mar 27. PMID: 35343091; PMCID: PMC9043727.

Question 2:Editing by an authority is required.  That is not my job as a reviewer other than to point out that most of the problems are associated with the use of an incorrect tense.

Reply:We tried our best to improve and make some changes to the manuscript. These changes will not influence the content and framework of the paper. And here we didn’t list the changes but marked in the revised paper. We appreciate for reviewers’ warm work earnestly and hope that the correction will meet with approval.

Reviewer 2 Report

Comments and Suggestions for Authors

In the manuscript “A single-Cell landscape of spermioteleosis in mouse and pig” the Authors provided data of transcriptomic analysis of single-cells from testis (including germ cells) and epididymis (without sperm cells). These result might be potentially interesting for readers working with spermatogenesis but the manuscript has some flaws that should be addressed before it can be accepted.

First of all, I haven’t found the reason for comparing mouse and pig transcriptomes. Why these two species ? What the Authors wanted to check or verify ? They only state “However, a comprehensive understanding of spermatogenesis and sperm maturation across mouse and pig lineages are still lacking.” That is far not enough, many other comprehensive comparisons between dozens of species are still missing, should all of them be done ? I am sure it was the Authors intention. But what was then ?

Second, the supplementary tables are missing. I assume they contain the data of particular genes from transcriptomic analysis. Without access to these data it is hard to review this manuscript. I have tried to get the data from National Genomic Data Center under provided GSA number, but it returns 0 hits.

And third, L88-95 The generation of sperm-depleted single cell suspension of epididymidal regions should be described in more details as this is critical step of sample preparation. How authors determined that epididymis are free of sperm ? The cited ref. 7 does not provide this information.

Minor concerns:

L35-37 Comprising three primary regions: the caput, corpus, and cauda, epididymis witnesses the transition of non-motile sperm, entry from the testis, to mature, motile sperm with fertilization capacity - actually sperm are able to fertilize the oocyte after capacitation in female tract. This sentence should be corrected.

L53-55 Numerous studies have identified the expression of innate immunity secretory genes in epididymal epithelial cells, playing crucial roles during spermatogenesis and maturation – spermatogenesis takes place in the testis, this should be corrected

L74-75 Sample collection was conducted under license – the license number should be provided. Also for pig experiments the license number should be provided. Why again the information about the ethical statement is given in L185-187?

L171 – it is fluorescent immunostaining not fluorescence in situ hybridization as the last term (known as FISH) is completely different technique

L535 The Authors defined Leydig cells cluster within epididymidal somatic cells ? This should be explained.

L618-619 This suggests that spermatogenesis in testis is a continuous process which different from the process of sperm maturation. – well, it is known for decades that spermatogenesis is a continuous process. Either the Authors wanted to state something else, in that case the sentence should be rephrased, or it can be omitted.

Comments on the Quality of English Language

English language should be improved.

Author Response

 In the manuscript “A single-Cell landscape of spermioteleosis in mouse and pig” the Authors provided data of transcriptomic analysis of single-cells from testis (including germ cells) and epididymis (without sperm cells). These result might be potentially interesting for readers working with spermatogenesis but the manuscript has some flaws that should be addressed before it can be accepted.

Question 1:First of all, I haven’t found the reason for comparing mouse and pig transcriptomes. Why these two species ? What the Authors wanted to check or verify ? They only state “However, a comprehensive understanding of spermatogenesis and sperm maturation across mouse and pig lineages are still lacking.” That is far not enough, many other comprehensive comparisons between dozens of species are still missing, should all of them be done ? I am sure it was the Authors intention. But what was then ?

Reply:Thank you very much for your valuable advice. Our explanation of the problem has been added to the original manuscript in line 56 to line 64. The details are as follows: “Animal models play a key role in scientific research. Mouse has high degree of genetic purity as well as high fertility. Therefore, mouse is the commonly used model in mammal related research [1]. As rodents, mouse is different from domestic animals to some extent. Therefore, we induced pig as a contrast model. First, pig is the most common domestic animal in China. Besides, pig is more human-like, in both of anatomy and physiology, than mice, making them another attractive option for simulating human [2].Therefore, we used mouse as model animals to elucidate the mechanism of spermatogenesis and maturation and adopted pig as the livestock model to further study spermatogenesis and maturation.”

[1]Rosenthal N, Brown S. The mouse ascending: perspectives for human-disease models. Nat Cell Biol. 2007 Sep;9(9):993-9. doi: 10.1038/ncb437. PMID: 17762889.

[2]Hou N, Du X, Wu S. Advances in pig models of human diseases. Animal Model Exp Med. 2022 Apr;5(2):141-152. doi: 10.1002/ame2.12223. Epub 2022 Mar 27. PMID: 35343091; PMCID: PMC9043727.

Question2:Second, the supplementary tables are missing.I assume they contain the data of particular genes from transcriptomic analysis. Without access to these data it is hard to review this manuscript. I have tried to get the data from National Genomic Data Center under provided GSA number, but it returns 0 hits.

Reply:We are sorry for our careless mistake. We have re-uploaded all the tables. Besides, we have sent the data to National Genomic Data Center and the GSA number is GSE249819. At present, we have released the relevant data.

Question3:And third, L88-95 The generation of sperm-depleted single cell suspension of epididymidal regions should be described in more details as this is critical step of sample preparation. How authors determined that epididymis are free of sperm ? The cited ref. 7 does not provide this information.

Reply:Thank you very much for pointing it out for us. The relevant instructions from line 106 to line 110 are as follows: “Once the dissected samples were cleared of any visible sperm, 3 mL of the sample mixture was transferred to a small 25 mL glass Erlenmeyer flask containing 7 mL of dissociation media (Collagenase IV 4 mg/ mL, Deoxyribonuclease I (DNAse I) 0.05 mg/mL) (Servicebio, China) and placed in a 35ËšC water bath with rotations of 200 rpm for 30 to 45 minutes.” This step would remove most of the sperm. We have deleted the sentence “Tissue dissociation was optimized to obtain a single cell suspension free of sperm.”

The cited ref. 7 shows that the epididymis consists of three parts: caput, corpus, and cauda, as well as how to divide the three parts. This reference is consisting with the content in original manuscript.

Question4:L35-37 Comprising three primary regions: the caput, corpus, and cauda, epididymis witnesses the transition of non-motile sperm, entry from the testis, to mature, motile sperm with fertilization capacity - actually sperm are able to fertilize the oocyte after capacitation in female tract. This sentence should be corrected.

Reply:Thank you for your advice. We changed the original content to the following sentence: Comprising three primary regions: the caput, corpus, and cauda, the epididymis facilitates further modifications of non-motile sperm.

Question5:L53-55 Numerous studies have identified the expression of innate immunity secretory genes in epididymal epithelial cells, playing crucial roles during spermatogenesis and maturation – spermatogenesis takes place in the testis, this should be corrected

Reply:Thank you for your advice. We changed the original content to the following sentence: “Numerous studies have identified the expression of innate immunity secretory genes in epididymal epithelial cells, playing crucial roles during sperm maturation.”

Question6:L74-75 Sample collection was conducted under license – the license number should be provided. Also for pig experiments the license number should be provided. Why again the information about the ethical statement is given in L185-187?

Reply:Thank you very much for pointing out the problem. The license number of pig had been provided from line 97 to line 100: “Six 12-month-old Landrace pigs were obtained from Haiyang Hexing animal husbandry Co., Ltd (no. 11002009000017, production licence number: SCXK: 2023-9065, Yantai, China)” Besides, we deleted this sentence that seemed to be repeated as the ethical statement.

Question7:L171 – it is fluorescent immunostaining not fluorescence in situ hybridization as the last term (known as FISH) is completely different technique

Reply:Thank you very much for pointing out the problem. It is absolutely that fluorescent immunostaining and fluorescence in situ hybridization is completely different technique. However, in this study ,we employed both fluorescent immunostaining and fluorescence in situ hybridization technology. One is ELEVL2 and CCNB2 immunofluorescent staining experiments on mouse and pig testis. The other is Defb20 and Defb30 fluorescence in situ hybridization on mouse epididymis.

Question8:L535 The Authors defined Leydig cells cluster within epididymidal somatic cells ? This should be explained.

Reply:Thank you very much for your valuable advice.These cells have high expression of genes associated with androgens such as Insl3,Cyp11a1 and Star. These genes are marker genes for Leydig cells. Besides, this cells also highly expressed epithelial marker genes such as Col1a1. Therefore, in the manuscript, this cell type had been modified to the epithelial cells. All relevant parts of the manuscript and images have been modified.

Question9:L618-619 This suggests that spermatogenesis in testis is a continuous process which different from the process of sperm maturation. – well, it is known for decades that spermatogenesis is a continuous process. Either the Authors wanted to state something else, in that case the sentence should be rephrased, or it can be omitted.

Reply:Thank you very much for your constructive suggestions. We omitted the sentence “This suggests that spermatogenesis in testis is a continuous process which different from the process of sperm maturation.” in the manuscript.

Reviewer 3 Report

Comments and Suggestions for Authors

Zhang et al. performed single-cell RNA-seq analyses with the testis and epididymis from mouse and pig. They classified those cells into clusters and subclusters, confirmed the expression of marker genes in each cluster/subcluster, described the difference and similarity between mouse and pig, and determined the localization of some marker genes. Overall, the manuscript seems to provide solid data. However, because similar data were published elsewhere, the authors should clarify the merit to publish these data, and the manuscript should be better written.

-English must be edited by a native speaker familiar with this field or a professional company.

-Some of their supplemental tables and figures are missing.

-Letters in some figures are too small.

-Check all references. For example, in page 11, lines 443-444, reference 47 was the study with mouse not pig.

-Why did the authors focus on mouse and pig? Single-cell RNA-seq data of mouse and pig testis and epididymis were published elsewhere (e.g., Journal of Animal Science and Biotechnology 12:122, 2021; Journal of Genetics and Genomics 49:1016, 2022; Cell Research 28:879, 2018; Cells 11:4135, 2022; Nature Communications 14:2499, 2023; eLife 9:e55474, 2020), and what is the significance of this study? The introduction should mention those previous data and justify the purpose of this study. The discussion should include the comparison of the current data with those published before.

-While the authors identified testicular cell types in detail by single-cell RNA-seq, their IHC/ISH data did not describe such detailed cell types. For example, what stages of spermatocytes were stained by the CCNB2 antibody? In relation to this, higher resolution of photos should be shown for IHC/ISH data.

-In section 2.2.2., did the authors remove Leydig cells? For pig, did they use the entire testis or a part of testis? The population of various cell types might change depending on the position of the testis piece.

-In section 2.2.4., what was the percentage of “high-quality cells”?

-In section 2.4., describe the details of antibodies.

-Section 2.5. seems to describe the method for immunohistochemistry not in situ hybridization.

-The sentence in page 16, lines 593-594 is the overstatement.

Comments on the Quality of English Language

-English must be edited by a native speaker familiar with this field or a professional company.

Author Response

Zhang et al. performed single-cell RNA-seq analyses with the testis and epididymis from mouse and pig. They classified those cells into clusters and subclusters, confirmed the expression of marker genes in each cluster/subcluster, described the difference and similarity between mouse and pig, and determined the localization of some marker genes. Overall, the manuscript seems to provide solid data. However, because similar data were published elsewhere, the authors should clarify the merit to publish these data, and the manuscript should be better written.

Question 1: -English must be edited by a native speaker familiar with this field or a professional company.

Reply:Thank you very much for your valuable advice, we have invited a native speaker to modify the manuscript. These changes will not influence the content and framework of the paper. And here we didn’t list the changes but marked in the revised paper. We appreciate for reviewers’ warm work earnestly and hope that the correction will meet with approval.

Question2:-Some of their supplemental tables and figures are missing.

Reply:Thank you for pointing out the errors for us. We are sorry for our careless mistake. The supplemental tables and figures have been reloaded in the supplemental data.

Question3:-Letters in some figures are too small.

Reply:Thank you very much for your valuable advice. According to your suggestion, we have modified the letters larger.

Question4:-Check all references. For example, in page 11, lines 443-444, reference 47 was the study with mouse not pig.

Reply:Thank you very much for your valuable advice. The reference 47 was the study with mouse just identified our mouse data in figure S7.We modified the details at line 493-494 as follows:The spermatids marker genes Acrv1 were highly expressed in mouse spermatocytes developmental trajectory state 4 (Figure S7A)

Question5-Why did the authors focus on mouse and pig? Single-cell RNA-seq data of mouse and pig testis and epididymis were published elsewhere (e.g., Journal of Animal Science and Biotechnology 12:122, 2021; Journal of Genetics and Genomics 49:1016, 2022; Cell Research 28:879, 2018; Cells 11:4135, 2022; Nature Communications 14:2499, 2023; eLife 9:e55474, 2020 , and what is the significance of this study? The introduction should mention those previous data and justify the purpose of this study. The discussion should include the comparison of the current data with those published before.

Reply:Thank you very much for your constructive suggestions. We have studied the single-cell RNA-seq data published elsewhere, and list the details as follows: 1. This paper focuses on the differentiation of spermatogonial cells and identifies four distinct subgroups. Journal of Animal Science and Biotechnology 12:122, 2021; 2.The study primarily investigates the somatic cell changes that occur during spermatogenesis in pig testis. Journal of Genetics and Genomics 49:1016, 2022; 3. This article provides a concise overview of mouse spermatogenesis with emphasis on the theory of single-cell sequencing. Cell Research 28:879, 2018; 4. The research examines the impact of specific gene expression on spermatogenesis in mice. Cells 11:4135, 2022; 5. This article extensively discusses the role of DDX4 gene in regulating spermatogenesis. Nature Communications 14:2499, 2023; 6. In this paper, single cell sequencing of mouse epididymis was done, and the spermatogenesis process was not elaborated. eLife 9:e55474, 2020.

The modification of the advice has been added to the introduction in line 50 to line 54. The details are as follows: “Previous studies mainly provided a concise overview and specific gene expression on mouse spermatogenesis and sperm mutation. Besides, there was report on the somatic cell changes that occur during spermatogenesis in pig testis . In contrast to previous studies that solely focused on either spermatogenesis or sperm maturation, the study comprehensively explores both processes.”

 In the discussion, we added the comparison of the current data with those published before at line 786-791. Details are as follows: In contrast to previous single-cell analyses on mouse[1-4] and pig[5], the current data provides a comprehensive exposition elucidating the dynamic modifications occurring on sperm from the testis to the epididymis. Expanding upon prior research, the study offers an intricate examination of gene expression during spermatogenesis and sperm maturation in mice and pigs, while also identifying novel marker genes.”

[1] Tan H, Wang W, Zhou C, Wang Y, Zhang S, Yang P, Guo R, Chen W, Zhang J, Ye L, Cui Y, Ni T, Zheng K. Single-cell RNA-seq uncovers dynamic processes orchestrated by RNA-binding protein DDX43 in chromatin remodeling during spermiogenesis. Nat Commun. 2023 Apr 29;14(1):2499. doi: 10.1038/s41467-023-38199-w. PMID: 37120627; PMCID: PMC10294715.

[2] Lu J, Liao J, Qin M, Li H, Zhang Q, Chen Y, Cheng J. Single-Cell RNAseq Resolve the Potential Effects of LanCL1 Gene in the Mouse Testis. Cells. 2022 Dec 19;11(24):4135. doi: 10.3390/cells11244135. PMID: 36552898; PMCID: PMC9777014.

[3] Chen Y, Zheng Y, Gao Y, Lin Z, Yang S, Wang T, Wang Q, Xie N, Hua R, Liu M, Sha J, Griswold MD, Li J, Tang F, Tong MH. Single-cell RNA-seq uncovers dynamic processes and critical regulators in mouse spermatogenesis. Cell Res. 2018 Sep;28(9):879-896. doi: 10.1038/s41422-018-0074-y. Epub 2018 Jul 30. PMID: 30061742; PMCID: PMC6123400.

[4] Rinaldi VD, Donnard E, Gellatly K, Rasmussen M, Kucukural A, Yukselen O, Garber M, Sharma U, Rando OJ. An atlas of cell types in the mouse epididymis and vas deferens. Elife. 2020 Jul 30;9:e55474. doi: 10.7554/eLife.55474. PMID: 32729827; PMCID: PMC7426093.

[5] Zhang L, Li F, Lei P, Guo M, Liu R, Wang L, Yu T, Lv Y, Zhang T, Zeng W, Lu H, Zheng Y. Single-cell RNA-sequencing reveals the dynamic process and novel markers in porcine spermatogenesis. J Anim Sci Biotechnol. 2021 Dec 7;12(1):122. doi: 10.1186/s40104-021-00638-3. PMID: 34872612; PMCID: PMC8650533.

Question6:-While the authors identified testicular cell types in detail by single-cell RNA-seq, their IHC/ISH data did not describe such detailed cell types. For example, what stages of spermatocytes were stained by the CCNB2 antibody?  In relation to this, higher resolution of photos should be shown for IHC/ISH data.

Reply:We are sorry that related antibodies can identify cell types but are not sufficient to distinguish cell states. Besides, the spermatogonocytes in this paper are not identified by stages. What’ more, we had uploaded a higher resolution image for ISH data.

Question7:-In section 2.2.2., did the authors remove Leydig cells? For pig, did they use the entire testis or a part of testis? The population of various cell types might change depending on the position of the testis piece.

Reply:Thank you very much for your valuable suggestion. We didn’t remove the Leydig cells in single-cell seq. For pig and mouse, the same part of the testicular tissue were used in single cell sequencing. We corrected the methods of testis cell suspensions generation at line 131 to line 138. The details are as follows: “The testis from mouse and pig were cleaned and washed with PBS. These samples were stored in MACS Tissue Storage Solution within 48 h (Miltenyi, Germany). Before dissociation, the testis tissues were cut into small pieces and transferred to 0.2% collagenase IV and DNase I digestion solution, followed by incubation at 37 °C for 15 minutes. After digestion and mechanical striking into single cells, the cell suspension was filtered through a cell strainer and converted to barcoded scRNA-seq libraries in accordance with the manufacturer’s protocol. The sequencing libraries were sequenced by DNBSEQ-T7.”

Question8:-In section 2.2.4., what was the percentage of “high-quality cells”?

Reply:Thank you very much for your valuable question. We feel sorry that we did not calculate the relevant proportions. Briefly, we removed cells with less than 200 detected genes and genes detected in 3 or fewer cells.

Question9:-In section 2.4., describe the details of antibodies.

Reply:Thank you very much for your valuable suggestions.The manufacturer and item number of the relevant antibody are as follows: CCNB2 Cyclin B2 Rabbit mAb (A7956, ABclonal, China), ELAVL2 Rabbit pAb (A5918, ABclonal, China).The related details ware added at manuscript in line 163-164.

Question10:-Section 2.5. seems to describe the method for immunohistochemistry not in situ hybridization.

Reply:Thank you again for your valuable advice. According to your suggestion, we have corrected the relevant experimental methods,detail as follows at line 175 to line 189: “Mouse epididymis slices were baked in an oven at 62°C for 2 hours, stored in xylene for 30 minutes to dewax, then treated with absolute ethanol for 20 minutes and soaked in DEPC water. The slices were repaired in citric acid antigen retrieval solution at 96°C for 10 minutes and cooled naturally to room temperature. Proteinase K (20ug/ml) (Servicebio,China) was added dropwise, digested at 37°C for 10 minutes, washed 3 times with PBS. Add the prehybridization solution dropwise and incubate at 37°C for 1h. Add hybridization solution containing probe dropwise, incubated at 37°C and washed with saline sodium citrate (SSC) (Servicebio,China).Incubation primary antibody: drop plus primary antibody, diluted with PBS (Servicebio,China) overnight at 4°C. Then wash with PBS for 3×5 minutes. Incubation of secondary antibodies: Add corresponding secondary antibodies and incubate at room temperature for 50mintes. Then wash with PBS for 3×5 minutes. Cell nuclei were labeled with 4′,6-diamino-2-phenylindole (DAPI) (Servicebio,China), observed and photographed under a laser scanning confocal microscope.

Question11:-The sentence in page 16, lines 593-594 is the overstatement.

Reply:Thank you very much for your valuable suggestions, we have deleted the relevant part.

Round 2

Reviewer 1 Report

Comments and Suggestions for Authors

The authors addressed the main concerns of this reviewer and made major improvements in the use of English in re-writing the manuscripts.  In particular they explained the basis for comparing pig and mouse spermatogenesis.  The data are now well described and likely to be of interest to the scientific community.

Comments on the Quality of English Language

Minor editing would be in order.

Author Response

Thank you for acknowledging our manuscript. We have thoroughly reviewed the manuscript and implemented necessary revisions. Thanks again for granting your approval.

Reviewer 2 Report

Comments and Suggestions for Authors

The Authors addressed all of my concerns.

Comments on the Quality of English Language

The Authors improved the mancuscript

Author Response

(The authors gave the same response as above.)

Reviewer 3 Report

Comments and Suggestions for Authors

The manuscript is improved very much, but the following points need to be further addressed.

-The method of in situ hybridization is still confusing. The authors hybridized epididymal section with a probe and treated the section with primary antibody, and the membrane (?) was incubated with the secondary antibody? They must explain what probe they used, how they prepared/labelled the probe, and what antibodies they used.

-Figure 4D shows some positive signals in spermatocytes at early stages in pig, but some epithelial cells and spermatogonia seems to be also stained. From this aspect, the description like ‘the CCNB2 protein was expressed in pig spermatocytes’ is not accurate. The authors should amend the descriptions concerning this point.

-In section 3.8., the description ‘This suggests that sperm motility begins in the late stages of spermatocyte development’ is not accurate. The sperm motility is not acquired in the testis.

-In section 3.10., in the first paragraph, “(Figure 8C-8G)” must be “(Figure 7C-7G)”.

Author Response

Thank you for your comments concerning our manuscript entitled “A Single-cell Landscape of Spermioteleosis in Mouse and Pig”. The comments are very helpful for improving our manuscript. We have studied comments carefully again and have made corrections which we hope to meet with your approval. The main corrections in the manuscript and the responds to the reviewers’ comments are as follows:

Question 1: The method of in situ hybridization is still confusing. The authors hybridized epididymal section with a probe and treated the section with primary antibody, and the membrane (?) was incubated with the secondary antibody? They must explain what probe they used, how they prepared/labelled the probe, and what antibodies they used.

Reply: Thank you very much for your kind words about our work. We have optimized the method of in situ hybirdization. Details as follows: The mouse epididymal paraffin section was incubated at 62℃ for 2 hours, followed by xylene dewaxing for 30 minutes. Subsequently, the section was treated with anhydrous ethanol for 20 minutes and washed with PBS for 10 minutes. The slices were repaired using citric acid antigen retrieval solution (ServiceBio, China) at a temperature of 96°C for 10 min and allowed to cool naturally to room temperature. A digestion step was performed by adding protease K (ServiceBio, China) at a concentration of 20µg/ml in a volume of 100 μl, followed by incubation at 37℃ for 10 minutes and subsequent washing with PBS three times. Prehybridization solution (100 μl) was added and incubated at a temperature of 37°C for 1 hour. Then, hybridization solution containing the RNA probe Defb20 (5‘UTR: CATCTGAGTGCCAAAGTTCTAAAACATCTTGGGCTGCTTAACATCTGGGCTCTAATCTGGCCTAATCTGCTTCTTCAG), Defb30 (5‘UTR: AAGAGCACGAGGGTCAACTGGCACTGGTAGGGAGGAGAGCAGCAGGTGTAAATCCGTTTTTTCATGTGACTGATG) was added (60μl). Incubation overnight took place at a temperature of 37 °C followed by washing with saline sodium citrate (SSC). For signal detection, the hybrid solution containing the signal probe (ServiceBio, China) was added in a volume of 60 μl and incubated at a temperature of 37 ℃ for 1 hour before being washed again with SSC. Finally, nuclei were labeled using 4',6-diamino-2-phenylindole (DAPI; Beyotime, China), observed under laser scanning confocal microscope (Hitachi, Japan).

Question 2: Figure 4D shows some positive signals in spermatocytes at early stages in pig, but some epithelial cells and spermatogonia seems to be also stained. From this aspect, the description like ‘the CCNB2 protein was expressed in pig spermatocytes’ is not accurate. The authors should amend the descriptions concerning this point.

Reply: Thank you very much for your valuable suggestion. We amended the descriptions concerning to CCNB2 immunostaining in pig, the details in section 3.8 are as follows: Our study demonstrated the expression of CCNB2 in early-stage spermatocytes and epithelial cells, as well as spermatogonia in pigs. Additionally, our findings revealed the presence of CCNB2 protein in Leydig cells in mice, which aligns with the developmental trajectory analysis. Therefore, CCNB2 could serve as a valuable marker for identifying Leydig cells in mice.

Question 3: In section 3.8, the description‘This suggests that sperm motility begins in the late stages of spermatocyte development’ is not accurate. The sperm motility is not acquired in the testis.

Reply: Thank you for your advice. We have deleted the sentence ‘This suggests that sperm motility begins in the late stages of spermatocyte development’in the manuscript.

Question 4: In section 3.10., in the first paragraph, “(Figure 8C-8G)” must be “(Figure 7C-7G)”.

Reply: Thank you very much for pointing out the problem. We had changed the ‘(Figure 8C-8G)’ to ‘(Figure 7C-7G)’.
